# MCT-Induced Ketosis and Fiber in Rheumatoid Arthritis (MIKARA)—Study Protocol and Primary Endpoint Results of the Double-Blind Randomized Controlled Intervention Study Indicating Effects on Disease Activity in RA Patients

**DOI:** 10.3390/nu15173719

**Published:** 2023-08-25

**Authors:** Christina Heidt, Jörn Pons-Kühnemann, Ulrike Kämmerer, Thorsten Marquardt, Monika Reuss-Borst

**Affiliations:** 1Faculty of Medicine, University of Muenster, 48149 Muenster, Germany; 2Department of General Pediatrics, Metabolic Diseases, University of Muenster, Albert-Schweitzer-Campus, 48149 Muenster, Germany; 3Medical Statistics, Institute of Medical Informatics, Justus Liebig University, 35392 Giessen, Germany; 4Department of Obstetrics and Gynaecology, University Hospital of Wuerzburg, 97080 Wuerzburg, Germany; 5Hescuro Clinics Bad Bocklet, 97708 Bad Bocklet, Germany; 6Department of Nephrology and Rheumatology, Georg-August University of Goettingen, 37075 Goettingen, Germany

**Keywords:** rheumatoid arthritis, medium-chain triglycerides, fiber, Simplified Disease Activity Index, β-hydroxybutyrate

## Abstract

Fatty acids, such as medium-chain fatty acids (MCFAs) and short-chain fatty acids (SCFAs), both important components of a normal diet, have been reported to play a role in bone-related diseases such as rheumatoid arthritis (RA). However, the role of medium-chain triglycerides (MCTs) has not been investigated in RA to date. The aim of this study was to investigate the effect of supplementation of regular diet with MCT with and without fiber on disease activity as measured with the SDAI (Simplified Disease Activity Index) in RA patients. A total of 61 RA patients on stable drug treatment were randomly assigned to a twice-daily control regimen or to a twice-daily regimen of a formulation containing medium-chain triglycerides (MCTs) 30 g/day for 8 weeks followed by a second twice-daily regimen of combining MCT (30 g/day) plus fiber (30 g/day) for an additional 8 weeks. The control group received a formulation containing long-chain triglycerides (LCTs) instead of MCTs. The preliminary results showed a significant reduction in SDAI from baseline to week 16 in the test group and a significant increase in β-hydroxybutyrate (BHB) levels, while no improvement in SDAI was observed in the control group.

## 1. Introduction

Rheumatoid arthritis is a worldwide chronic disease, characterized by joint swelling and pain and destructive changes in bone and cartilage of multiple joints, which if untreated, can lead to joint damage and disability [1,2,3]. The disease affects women 2–3 times more often than it does men [2]. To date, the pathogenesis of RA has not been fully understood, but genetic and environmental factors have been discussed as potential causes of RA [2,4]. Relevant environmental risk factors for RA are smoking and other airborne exposures (silica), as well as an unhealthy diet and microbiota [2,4,5]. The treatment of RA is multimodal and includes pharmacologic and nonpharmacologic therapies [6]. Apart from pharmacological treatment, patients are often interested in self-management strategies for symptomatic improvement such as diet. In recent years, an increasing number of studies have investigated the role of diet as possible adjunctive therapy for RA [7]. Specific foods (e.g., meat, sugar) could trigger RA or, on the contrary, reduce inflammation (fish, vegetables, fruits) [8]. Anti-inflammatory diets such as the Mediterranean diet or plant-based or ketogenic diets result in lower disease activity, pain, and cardiometabolic-related outcomes compared to ordinary diets [9,10,11,12].

The ketogenic diet (KD) is characterized by restricting the offset of carbohydrates using high-fat content to induce physiological ketosis through production of ketone bodies (e.g., BHB). The effect of KD on systemic inflammation is related among others to β-hydroxybutyrate (BHB) synthesis [12]. BHB has a direct anti-inflammatory effect on the NLRP_3_ inflammasome, which is a protein complex involved in monocyte-induced inflammation [13]. BHB reduces the expression of the NLRP_3_ (NOD-LRR- and pyrin domain-containing protein 3) inflammasome pathway (caspase 1) and also limits the release of pro-inflammatory cytokines (IL-1β and IL-18) [14,15]. Unfortunately, due to the marked restriction of carbohydrates, a KD can be very challenging and stressful [16].

Ketosis could be achieved via ketogenic diet or ingesting ketone precursor such as medium-chain fatty acids (MCFAs), caprylic acid (C8:0), and capric acid (C10:0) [15]. MCT supplementation can raise blood BHB up to 0.3 to 1.0 mmol/L because MCT degrades 5 to 8 times faster than does LCT from the gastrointestinal tract, and MCFAs are transported directly into the liver via the portal vein and rapidly metabolized to ketone bodies via conversion from acetyl CoA [15,17,18,19,20]. Exogenous ketosis from MCT is independent of the fasting state, plasma insulin or low to moderate carbohydrate intake and MCTs offer a potential advantage of inducing nutritional ketosis without the need for a drastic change in dietary habits [21]. Besides the benefits of ketosis, MCT can improve immunity, gut microbiota, and intestinal health [22,23]. 

Certain gut-associated metabolites including MCFAs and short-chain fatty acids (SCFAs) may play a role in bone-related diseases such as RA [24]. Many preclinical studies have confirmed that butyrate can ameliorate the damage in RA [25,26,27,28,29,30]. However, the role of MCTs has not been investigated in human RA studies to date. Thus, the aim of the present study was to investigate the effect of supplementation of a regular diet with MCT with and without fiber on disease activity in RA patients.

The primary research question was defined as a change in the Simple Disease Activity Index (SDAI) from baseline to week 16 in the test versus control group. Further examination of secondary research questions included changes in (1) metabolic blood profile, (2) gut microbiota, (3) physical function, and (4) quality of life.

Here, we report the study design, intervention, primary outcome, and the preliminary results based on the primary endpoint and secondary outcomes related to the primary endpoint from 56 RA patients.

## 2. Materials and Methods

### 2.1. Clinical Trial Design and Eligibility Criteria

The present trial was designed as a randomized, double-blind, placebo-controlled, 16-week, single-center study and was conducted between August 2021 and October 2022.

This study included RA patients who were under regular follow-up at the Rheumatology Outpatient Practice in Bad Bocklet, Germany. The inclusion criteria were (1) adult patients (>18–80 years), (2) with no evidence of metabolic disease other than obesity, (3) a diagnosis of rheumatoid arthritis fulfilling the RA classification criteria of the American College of Rheumatology/European League Against Rheumatism (ACR/EULAR), (4) and receiving stable treatment with disease-modifying antirheumatic drugs (DMARDs) (conventional synthetic, targeted synthetic, or biological DMARDs). Exclusion criteria from the study were (1) BMI ≥ 45 kg/m^2^; (2) antibiotic, prebiotic, and probiotic therapy up to 3 months before the study; (3) vegan diet, ketogenic diet, and MCT-rich diet (containing MCT, coconut, and or palm kernel oil); and (4) type 2 diabetes, cardiovascular disease, neurological and psychiatric disorders, and inflammatory bowel disease.

### 2.2. Randomization, Sample Size, and Intervention

RA patients were randomly assigned to either a test or control group with a ratio of 2:1 (f/m) by stratified randomization. The study flowchart is shown in Figure 1**.** Sample size was calculated for a one-sided Mann–Whitney U (Wilcoxon) test, with a 1.0-point difference in decrease for the SDAI, a standard deviation of 1.25, 90% power (d = 0.8), and a 5% level of significance (α) being considered for a sample size of 58 patients.

Interventions were either a formulation containing MCT (30 g/day) or a formulation containing LCT (30 g/day) for 8 weeks followed by a second twice-daily regimen combining MCT (30 g/day) plus fiber (30 g/day) for an additional 8 weeks or a twice-daily control regimen containing LCT (30 g/day) plus fiber (30 g/day). All four interventions were provided by Schär AG/SPA, Postal, Italy. The four products were provided as single-serving packets and had the same outer packaging (pouch) but different batch numbers to both blind and reliably assign the products. Each formulation was taken twice daily according to patient preference either as a porridge (with 25 to 50 mL of liquid prepared) or a drink (prepared with 50 to 100 mL of liquid). To avoid interfering with ketosis, the formulations were consumed with either drinking water or a carbohydrate-free milk alternative (defined by 0 g of carbohydrate and 0 g of sugar). Each formulation was taken 60 min before the usual diet (before breakfast and an afternoon snack). The undesirable gastrointestinal side effects of MCTs are not only dependent on the administration but also on the dosage [31]. Low doses between 10 g and 20 g of MCTs elicit the most gastrointestinal symptoms [32]. Moreover, for a ketogenic effect of <1 mmol/L lower doses are also sufficient [33,34]. Based on published data, the dosage of MCT was defined as 15 g per serving. Additionally, fiber, especially easily fermentable fiber, may cause undesirable gastrointestinal symptoms such as flatulence or diarrhea [35]. For reasons of better tolerability, >70% of fiber used in the test and control regimens was an insoluble fiber source. The dosage of 15 g of fiber per serving given twice daily was based on a previous clinical feasibility study of RA patients [36].

Consumption (day and time) was recorded twice a day for 16 weeks. The patients and investigators (rheumatologist, study nurse, and nutritionist) were blinded to the group allocation until the trial ended.

Patients were assessed at week 0 = T0 (baseline), week 8 = T2, and week 16 = T4 (see Figure 1). The specific assessments at each visit are detailed in Appendix A.

### 2.3. Study Outcomes

The difference in SDAI from baseline (T0) to 16 weeks (T4) was considered as the primary outcome of the study. Secondary outcomes were evaluated longitudinally from baseline (T0) to 8 weeks (T2) and to 16 weeks (T4) between the test and control groups for SDAI, CRP, VAS, and BHB. In addition, descriptive results at 3 time points (T0, T2, and T4) were provided for each endpoint. All study outcomes are presented in Appendix A**.**

### 2.4. Measurements

#### 2.4.1. Disease Activity

Disease activity was evaluated using the SDAI (Simple Disease Activity Index) as measured using the arithmetic sum of tender and swollen 28-joint count, the patient’s and rheumatologist’s global assessment, and CRP in mg/dL as described in Aletaha et al [37]. For CRP analysis, fasting blood samples from RA patients were drawn during the visits, and serum was separated and examined within <2 h after collection using the cobas c 502 analyzer (Roche Diagnostics, Mannheim, Germany). All standardized procedures were conducted by clinical laboratory staff at Hescuro Clinic Bad Bocklet, Germany. Serum hs-CRP concentrations were determined with particle-enhanced immunoturbidimetric assay (CRP4, tina-quant C-Reactive Protein IV).

#### 2.4.2. BHB Levels

Capillary BHB levels (mmol/L) were measured using GlucoMen^®^ areo 2K, a combined blood glucose and ß-ketone meter for home testing (A. Menarini Diagnostics S.r.l., Florence, Italy). Levels were recorded twice per day (a record template is available in Appendix A).

#### 2.4.3. Adherence

The patients were asked to record the consumption of the formulation (date and time) and the ketone levels twice daily. From both our own work and published data, measurable ketosis (BHB increase) is possible after 30 min [38]. The timing of BHB measurement was therefore set at 30 min after ingestion. To maintain compliance, telephone interviews were conducted weekly (or as needed) to clarify unanswered questions and check daily intake according to study protocol.

### 2.5. Ethical Approval and Trial Registration

This study has been reviewed and approved by the Bavarian Ethics Committee (approval number: 21020), and it was registered at the German Registry of Clinical Trials as DRKS00025413. Written informed consent was obtained from all patients before enrollment.

### 2.6. Statistical Analysis

Statistical analyses were performed using GraphPad Prism version 9.5.1 (GraphPad, La Jolla, CA, USA) and the R statistical programming language version 4.2.3 (R Core Team, 2023). Descriptive statistics were used to summarize demographic variables with median and interquartile range (IQR).

For the primary endpoint, the difference in SDAI defined as T0 minus T4 between the two groups were assessed using the Mann–Whitney U (Wilcoxon) test. In addition, group difference was proofed at each time point and longitudinally between time points within each group. Correlations between SDAI and VAS, CRP, and BHB were assessed using the Spearman correlation coefficient.

Linear mixed models (using the R LME4 package) were applied to evaluate the influence of time, intervention, and BHB levels on SDAI (dependent variable) [39]. The normality of residuals was examined using qq-plots and histograms. Because of a slight tendency to heteroscedasticity, robust estimators were applied.

## 3. Results

### 3.1. Patient Characteristics

To test the effect of MCT and fiber in RA patients, we performed a double-blinded, randomized, placebo-controlled intervention study and screened 80 adult patients diagnosed with RA. We recruited 61 patients with low to moderate disease activity as measured by SDAI. Of the 61 patients, 34 were randomly assigned at a ratio of 2:1 (f/m) to the test group and 27 to the control group. The two groups were well balanced in terms of gender (41 female and 20 male patients). Three female patients dropped out shortly after randomization (2 in the test group and 1 in the control group), and 58 patients completed visit 2 (T2) on week 8 (32 patients in the test group and 26 patients in the control group). At the end of the study, 56 patients (30 patients in the test group and 26 patients in the control group) completed visit 3 (T4) on week 16. With this sample size, the resulting test power still exceeded 89%.

The median age of all RA patients was 63 years, with a median disease duration of 2.5 years. Additionally, 34.4% were seropositive for rheumatoid factor (RF) and 27.9% for anticitrullinated protein antibodies (ACPAs). The clinical and demographic data of patients are presented in Table 1. The baseline anthropometric, metabolic, and nutrition data have been published previously [40,41]. Further patient data (history and medication) are presented in Appendix A.

### 3.2. Primary Outcome

Wilcoxon’s test indicated a significant difference in the decrease of the primary outcome SDAI in the control and test groups between T0 and T4 (*p* ≤ 0.05) (Figure 2).

Table 2 shows the difference in the SDAI defined as T0 minus T4, which displays the primary endpoint.

### 3.3. Secondary Outcomes

Analysis of secondary outcome variables included longitudinal results of SDAI, BHB, CRP, and VAS*patient* between the test and control groups at baseline, week 8, and week 16. Descriptive results are also shown for SDAI and BHB.

#### 3.3.1. SDAI

There was no significant difference between the two groups in the SDAI at the beginning of the study (T0). The SDAI decreased in both groups by week 8. There was a significant difference between the two groups at the end of the study (T4) (*p* = 0.03), as shown in Table 3.

The graphical demonstration of the descriptive changes in the SDAI is depicted in Figure 3. The median (IQR) SDAI decrease from T0 to T2 was 4.575 (1.525–8.210) in the test group and 0.565 (0.410–4.940) in the control group. Between T2 and T4, the SDAI decrease was 1.0550 (−0.0375–3.0075) in the test group, whereas in the control group, the SDAI was essentially unchanged (median −0.2400; IQR: −2.8900–3.6075).

#### 3.3.2. BHB, CRP, and VAS

Analysis of secondary outcome variables showed that serum BHB levels were significantly higher in T2 and T4 in a comparison of the test group with the control group. No significant difference was found for CRP or VAS*patient* at any of the 3 time points (Table 4). The VAS*patient* decreased in both groups by week 8 and only in the test group by week 16. The VAS*patient* in the control group was unchanged by week 16.

The graphical demonstration of the descriptive changes in BHB levels is shown in Figure 4. The median (IQR) BHB increase (*p* < 0.001) from T0 to T2 was 0.36 (0.26–0.51) in the test group and 0.06 (−0.00–0.11) in the control group. Between T2 and T4, the BHB increase (*p* = 0.09) was 0.03 (−0.02–0.08) in the test group, whereas in the control group, the BHB levels were significantly increased (*p* < 0.001) (median: −0.04, IQR: −0.08–0.00).

#### 3.3.3. Correlations between SDAI, BHB, CRP, and VAS

Correlations results between SDAI and BHB, CRP, and VAS*patient* are presented in Figure 5. The SDAI was positively associated with CRP (*p* < 0.001) and VAS (*p* < 0.001) and negatively correlated with BHB (*p* ≤ 0.05).

#### 3.3.4. Linear Mixed Models

Table 5 displays the linear mixed models results for the SDAI. Model 1 included only the intercept and the independent variables as the time point (TP), formulation (test group), and TP and formulation (test group). Model 2 included all independent variables, while Model 3 included significant interactions. The interpretation of results focused mainly on Model 3, which included all significant interactions. Significant interactions were found between TP and formulation (test group) (*p* < 0.01) and between BHB and TP (*p* ≤ 0.05). Generally, increasing BHB levels had a significantly negative effect on SDAI (β = −10.2, *p* < 0.01). From the interaction term, it could be concluded that the increase of BHB from T0 to T2 in the test group was associated with a decrease in the SDAI, but this association was not further observed from T2 to T4 due to stable BHB levels at week 16.

## 4. Discussion

This is the first randomized controlled study comparing MCT versus LCT supplementation alone and with fiber in patients with rheumatoid arthritis. The primary study outcome, the reduction in the SDAI by 5.23 points from baseline (T0) to 16 weeks (T4) in the test group consuming MCT alone and in combination with fiber, was highly significant compared to that in the LCT control group. More clinical confirmation is needed before this strategy could be recommended with confidence.

Improvement in disease activity has been reported for fasting, which increased ketone levels until the fast is broken with a meal containing significant carbohydrate [42,43]. Unfortunately, the results cannot be compared with results from this study because the results were either obtained partly before the era of modern drug therapy for RA or, more recently, from a fasting study followed by a plant-based diet (NutriFast), which showed no significant difference in the improvement of the SDAI in the intervention group compared with the control group [44]. The first RA human trial with dietary fiber also failed to show an improvement in disease activity in the intervention group [36]. A recently published randomized controlled clinical intervention trial (Plants for Joints) of 83 RA patients examined the effects of a multidisciplinary lifestyle program consisting of a plant-based diet, physical activity, and stress management, while the control group received standard therapy [10]. After 16 weeks of intervention, there was a 0.9-point, moderate improvement in Disease Activity Score 28 (DAS28) in the intervention group compared with the control group [10]. The authors also reported that fiber intake improved with the plant-based diet in the intervention group and contributed to meeting the recommended fiber intake [10].

In this study, the association between SDAI and interventions (MCT alone and with fiber) as well as BHB levels (alone and depending on the intervention) was investigated to provide evidence for a possible influence between these parameters. The result of the mixed linear regression analysis showed that decreasing SDAI was significantly associated with an increase in BHB levels. This effect was most strongly demonstrated in the test intervention from T0 to T2 and remained at a stable level with the addition of fiber and constant BHB concentrations from T2 to T4.

BHB can produce energy as ATP through the TCA cycle and act as an important signaling molecule in metabolic homeostasis and cell regulation [45,46]. BHB has also been reported to inhibit mTOR signaling pathways, IGF-1 and leptin, to enhance autophagy and to provide epigenetic effects through the inhibition of histone deacetylases [46,47,48]. BHB has been shown to play a role in mediating NLRP3 inflammasome-induced IL-1β and IL-18 in human monocytes to positively influence inflammation [13]. In addition, BHB has anti-inflammatory effects via an interaction with GPR109A and might, therefore, be useful in the treatment of inflammatory disorders [49,50].

Beneficial effects of omega-3 fatty acids (FA) on the clinical parameters of RA were presented in a recent systematic review of 71 studies [51]. Long-chain polyunsaturated FAs are precursors to eicosanoids, and those derived from omega-3 FAs exhibit anti-inflammatory properties [51]. Anti-inflammatory effects have also been described for MCTs [52,53,54]. Since studies comparing the effects of MCT versus long-chain unsaturated FAs in RA have not yet been reported, further studies comparing the effects of these FAs on disease activity in RA would be useful.

The results of this study show that MCTs induce a moderate increase in BHB after 30 min compared with LCTs. This is consistent with a large body of other work [34,38,55,56]. Studies also show that SCFAs, especially butyrate, can promote BHB production [57,58]. In this study, dietary fiber was not found to have a ketosis-inducing effect at T4 in either group. This could be the result of poor or delayed fermentation of the bamboo fiber such that the production of SCFAs and resulting ketosis occurred after the 30 min timeframe chosen to assess BHB levels [59,60]. Recently published data have shown for the first time, that specific fibers, such as bamboo fiber can promote the production of total SCFAs after 24 h of fermentation [60]. Future studies need to investigate this new information.

Oral high-dose MCT supplementation is associated with gastrointestinal distress including diarrhea, vomiting, and abdominal cramps [15,38,61]. As result of the chosen combination of MCT with low fermentable fiber source in this study, no RA patients reported any gastrointestinal symptoms. In agreement with our results, a recent study investigated the effects of high fiber multigrain supplementation (containing 15.6 g fiber) for 12 weeks on disease activity scores in patients with moderate to severe RA coupled with the prescribed antirheumatic regimen using a control group [62]. The multigrain supplementation group showed significant improvement in the DAS-28 score [62]. The authors also reported no dropouts [62].

The strength of this study is its double-blinded and randomized design with a power of 89% (n = 56) to 90% (n = 58) and well-balanced gender-based groups. Furthermore, RA patients were familiar to the rheumatologist, for most of the patient’s receive long-term treatment and are, therefore, very compliant with the interventions. However, this study has certain limitations. First, the research question of the primary endpoint, the reduction of the SDAI before (T0) and after (T4) was based on one dietary intervention: MCT and dietary fiber. The SDAI, a validated instrument developed by the EULAR, was used to assess disease activity because it provides a good discrimination between low, medium, and high disease activity and, in particular, remission compared to the DAS28 [63]. In addition, the assessment also incorporated CRP as an inflammatory parameter, which is not included in the Clinical Disease Activity Index (CDAI). Although this allowed for a better comparison of treatment effects and treatment options on disease activity, CDAI is recommended as the measurement tool of choice [64]. The experience in this study has shown that the subjective assessment of disease activity by the patient (VAS*patient*) can also be influenced by the patient’s overall state of health and may not always be associated with the inflammatory disease.

## 5. Conclusions

Our preliminary results showed a significant reduction in disease activity as measured with the SDAI, a nonsignificant but clear improvement in VAS*patient*, and a significant increase in BHB levels in RA patients through supplementation of the regular diet with MCT alone and in combination with fiber as compared to consuming a control supplement. We are currently investigating our hypothesis that the mechanism by which MCTs can further reduce disease activity is through microbiota-mediated host effects on gut-barrier function and autoimmunity.

## Figures and Tables

**Figure 1 nutrients-15-03719-f001:**
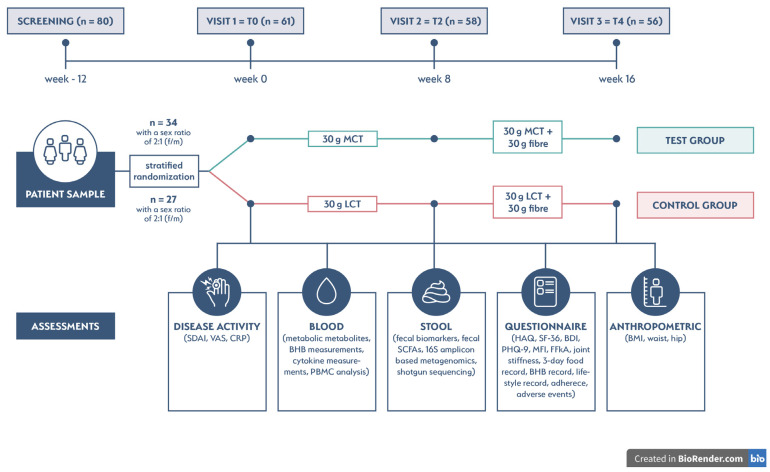
Study flowchart. (SDAI—Simplified Disease Activity Index; VAS—visual analogue scale, CRP—C-reactive protein, BHB—β-hydroxybutyrate; PBMC—peripheral blood mononuclear cell; SCFA—short-chain fatty acid; HAQ—health assessment questionnaire; SF-36—short form health survey; BDI—Beck-Depression Inventory; PHQ-9—Patient Health Questionnaire; MFI—Multidimensional Fatigue Inventory; FFkA—German questionnaire to assess health-related physical activity; BMI—body mass index).

**Figure 2 nutrients-15-03719-f002:**
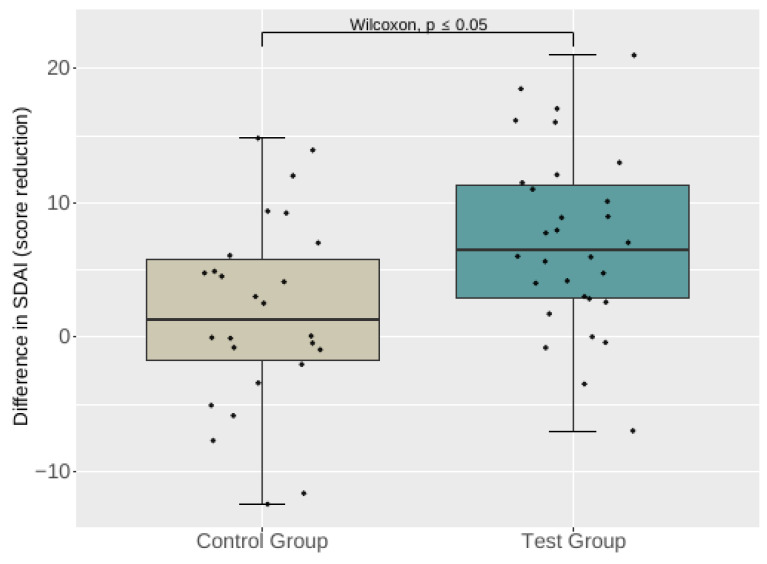
Primary endpoint: difference in the SDAI in the SDAI in the test and control groups between T0 and T4.

**Figure 3 nutrients-15-03719-f003:**
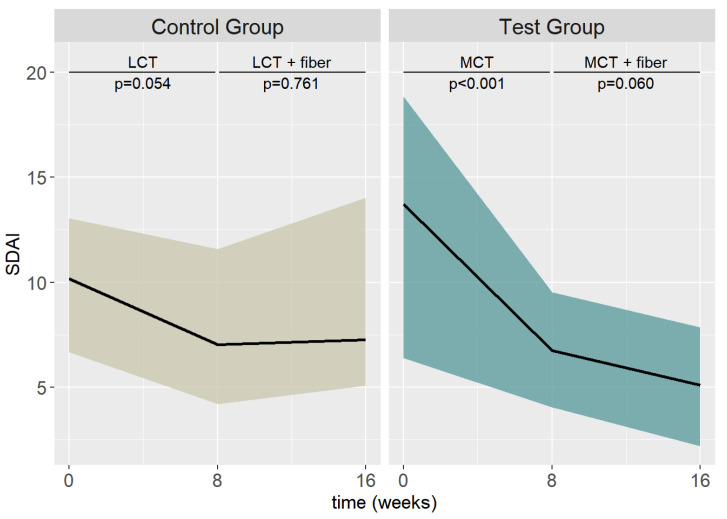
Descriptive SDAI results between the test and control group and T0, T2, and T4. (Values are shown as the median and IQR).

**Figure 4 nutrients-15-03719-f004:**
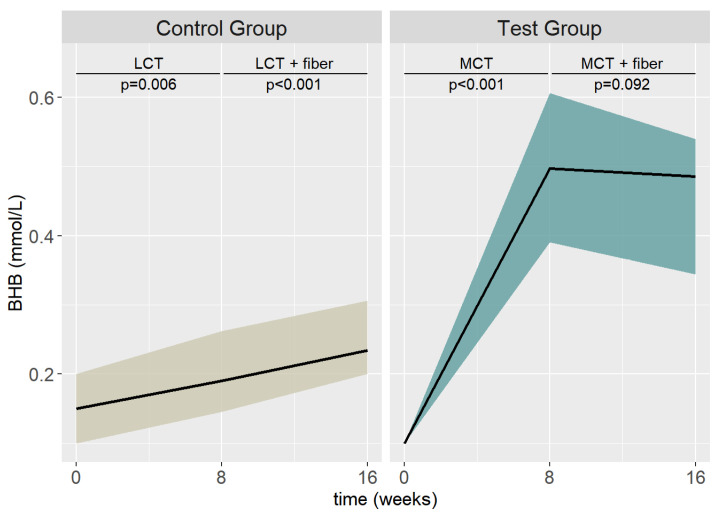
Descriptive BHB results between the test and control group at T0, T2, and T4. (Values are shown as the median and IQR).

**Figure 5 nutrients-15-03719-f005:**
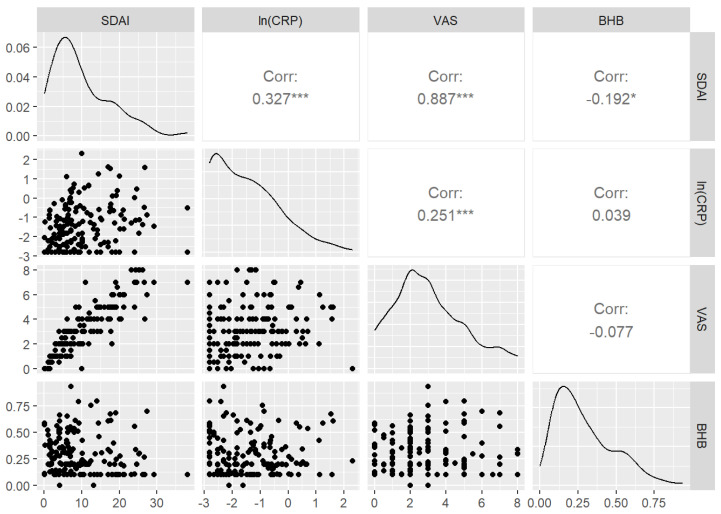
Correlations between SDAI and BHB, CRP, and VAS*patient*. Density of the distributions are shown in the diagonal. (Corr: Spearman correlation coefficient; ln: natural logarithm; * *p* ≤ 0.05; *** *p* < 0.001).

**Table 1 nutrients-15-03719-t001:** Baseline characteristics.

Variable	Control Group	Test Group
n	27	34
Female, n (%)	18 (67%)	23 (68%)
Age, years (median, IQR)	63.5 (56.2–70.8)	60.5 (56–70)
Disease duration, years (median, IQR)	3.5 (1.5–17)	1.3 (0.5–5.3)
IgM-RF positive, n (%)	9 (33)	12 (35)
ACPA positive, n (%)	8 (30)	9 (27)
SDAI, units (median, IQR)	10.16 (6.89–13.60)	13.72 (6.83–18.36)
CRP, mg/dL (median, IQR)	0.2 (0.1–0.57)	0.2 (0.07–0.44)
Pain (VAS), score (median, IQR)	3 (2–4)	3.5 (2–5)
Methotrexate, n (%)	8 (30)	11 (32)
Other (cs)-DMARDs, n (%)	3 (11)	4 (18)
(ts)-DMARDs, n (%)	1 (4)	2 (6)
Biologicals, n (%)	8 (30)	5 (15)
Glucocorticoids, n (%)	7 (26)	9 (27)

**Table 2 nutrients-15-03719-t002:** Difference in the SDAI defined as T0 minus T4.

N	Control Group	Test Group	*p*-Value
56	1.28 (−1.78–5.76) ^1^	6.51 (2.87–11.35) ^1^	<0.05
	1.75 ± 7.11 ^2^	7.18 ± 6.66 ^2^	

^1^ presented as median and IQR. ^2^ presented as mean ± standard deviation.

**Table 3 nutrients-15-03719-t003:** Longitudinal results of the SDAI between the two groups.

SDAI Units	Control Group	Test Group	*p*-Value
T0 (n = 61)	10.16 (6.69–13.04) ^1^	13.72 (6.38–18.82) ^1^	0.24
	11.36 ± 7.80 ^2^	13.91 ± 8.91 ^2^	
T2 (n = 58)	7.04 (4.21–11.57) ^1^	6.76 (4.06–9.53) ^1^	0.57
	9.48 ± 7.52 ^2^	8.14 ± 6.03 ^2^	
T4 (n = 56)	7.27 (5.08–14.02) ^1^	5.12 (2.21–7.85) ^1^	**0.03**
	9.84 ± 7.17 ^2^	6.38 ± 5.94 ^2^	

^1^ presented as median and IQR. ^2^ presented as mean ± standard deviation.

**Table 4 nutrients-15-03719-t004:** Longitudinal results of BHB, CRP, and VAS between the two groups.

BHB (mmol/L)	Control Group	Test Group	*p*-Value
T0 (n = 61)	0.15 (0.10–0.20) ^1^	0.10 (0.10–0.10) ^1^	**0.04**
	0.15 ± 0.06 ^2^	0.12 ± 0.05 ^2^	
T2 (n = 58)	0.19 (0.15–0.26) ^1^	0.50 (0.39–0.61) ^1^	**<0.001**
	0.20 ± 0.07 ^2^	0.51 ± 0.15 ^2^	
T4 (n = 56)	0.23 (0.2–0.31) ^1^	0.49 (0.34–0.54) ^1^	**<0.001**
	0.26 ± 0.10 ^2^	0.47 ± 0.14 ^2^	
**CRP (mg/dL)**			
T0 (n = 61)	0.16 (0.11–0.57) ^1^	0.19 (0.06–0.40) ^1^	0.47
	0.58 ± 1.0 ^2^	0.38 ± 0.59 ^2^	
T2 (n = 58)	0.20 (0.11–0.64) ^1^	0.17 (0.09–0.41) ^1^	0.55
	0.38 ± 0.36 ^2^	0.58 ± 1.18 ^2^	
T4 (n = 56)	0.17 (0.09–0.52) ^1^	0.21 (0.07–0.36) ^1^	0.74
	0.74 ± 1.94 ^2^	0.49 ± 0.85 ^2^	
**VAS*patient* (cm)**			
T0 (n = 61)	3.0 (2.0–4.0) ^1^	3.5 (2.0–5.0) ^1^	0.73
	3.5 ± 2.2 ^2^	3.6 ± 1.9 ^2^	
T2 (n = 58)	2.5 (1.6–4.9) ^1^	2.8 (1.9–3.3) ^1^	0.80
	3.0 ± 2.2 ^2^	2.8 ± 1.8 ^2^	
T4 (n = 56)	2.5 (2.0–4.0) ^1^	2.0 (1.0–3.0) ^1^	0.34
	2.9 ± 2.1 ^2^	2.4 ± 1.8 ^2^	

^1^ presented as median and IQR. ^2^ presented as mean ± standard deviation.

**Table 5 nutrients-15-03719-t005:** Three different regression models modeling the influence of time, intervention, and BHB levels on the dependent variable SDAI according to linear mixed models using robust estimators. Model 1 consists of only main effects; in Model 2, all two-way interactions were included, and in Model 3, only significant interactions remained.

Predictors		Model 1		Model 2		Model 3	
		Coefficient (β)	95% CI	Coefficient (β)	95% CI	Coefficient (β)	95% CI
Fixed effect	Intercept	**10.97 *****	8.02–13.93	**12.68 *****	8.52–16.84	**12.85 *****	9.63–16.08
	Time point (TP)	−0.41	−1.08–0.26	**−1.50 ****	−2.50–−0.50	**−1.49 ****	−2.53–−0.44
	formulation (test group)	2.37	−1.77–6.50	1.96	−2.89–6.81	1.78	−2.22–5.77
	TP and formulation (test group)	**−1.42 ****	−2.31–−0.52	**−1.72 ****	−2.86–0.58	**−1.78 ****	−2.81–0.74
	BHB			−9.00	−26.01–8.01	**−10.2 ****	−17.63–2.78
	BHB and TP			**5.11 ***	−1.17–9.06	**5.19 ****	−1.53–8.85
	BHB and formulation (test group)			−1.23	−17.37–14.91		
Random effect	σ^2^	11.38		9.92		9.86	
Marginal R^2^/conditional R^2^	R^2^	0.084/0.814		0.107/0.820		0.107/0.820	

* = significant at *p* < 0.05. ** = significant at *p* < 0.01. *** = significant at *p* < 0.001.

## Data Availability

Not applicable.

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
