# Peer review of "MCT-Induced Ketosis and Fiber in Rheumatoid Arthritis (MIKARA)—Study Protocol and Primary Endpoint Results of the Double-Blind Randomized Controlled Intervention Study Indicating Effects on Disease Activity in RA Patients"

_nutrients, 2023, doi:10.3390/nu15173719_

Round 1
Reviewer 1 Report
Abstract
Please define BHB (line 28)
In the Introduction is written that “61 RA patients on stable drug treatment were randomly assigned” (line 22). At the same time, in Materials and Methods, it stated “resulting in a sample size of 58 patients” (line 101). In addition, in the results, it is written that 56 patients completed the study. Please state in the Introduction (in the last sentence) that these results are from n=56.
Introduction
Line 51 – define BHB
Line 53 – define NLRP3
Line 61 – delete (without transporters)
Materials and methods
Please provide MCT, LCT, and fibers' source (company, country, and content).
Line 134 – change to C-reactive protein
Line 149 – Please add the name according to Aletaha et al. [42].
Figure 1 contains STOOL on week 8. Is the data presented?
Figure 1 contains QUESTIONNAIRE (HAQ, SF-36, MFI, etc). Is the data presented?
Figure 1 - Is the data of cytokine measurements presented?
Figure 1 - Is the data of PBMC analysis presented?
Results
Line 201 – define RF and ACPA
Table 1. Delete the % in 18 (67%)
Delete Table 2. It is already explained in the text.
Why the authors did not present the results from the secondary points from Table S2?
Please include the following papers and discuss their results too:
Farzana Athirah Abdul Latif, Wan Syamimee Wan Ghazali, Siti Mardhiana Mohamad, Lai Kuan Lee, High fiber multigrain supplementation improved disease activity score, circulating inflammatory and oxidative stress biomarkers in rheumatoid arthritis (RA) patients: A randomized human clinical trial, Journal of Functional Foods, Volume 100, 2023, 105392, https://doi.org/10.1016/j.jff.2022.105392.
Tański W, Świątoniowska-Lonc N, Tabin M, Jankowska-Polańska B. The Relationship between Fatty Acids and the Development, Course and Treatment of Rheumatoid Arthritis. Nutrients. 2022 Feb 28;14(5):1030. doi: 10.3390/nu14051030.
Author Response
Dear Reviewer,
First, thank you very much for this helpful comment, which is truly very important. We have included all your comments to our article. You will find the updated information highlighted in red.
Thank you for your second review in advance!
With very best wishes,
Christina and all Authors
_____________________________________________________________
Comments and Suggestions for Authors
Abstract
Please define BHB (line 28)
Response 1: we have added β-Hydroxybutyrate as definition.
In the Introduction is written that “61 RA patients on stable drug treatment were randomly assigned” (line 22). At the same time, in Materials and Methods, it stated “resulting in a sample size of 58 patients” (line 101). In addition, in the results, it is written that 56 patients completed the study. Please state in the Introduction (in the last sentence) that these results are from n=56.
Response 2: please see line 81.
Introduction
Line 51 – define BHB
Response 3: we have added β-Hydroxybutyrate as definition.
Line 53 – define NLRP3
Response 4: we have added the long-term “NOD-LRR- and pyrin domain containing protein 3” as definition.
Line 61 – delete (without transporters)
Response 5: we have deleted the phrase.
Materials and methods
Please provide MCT, LCT, and fibers' source (company, country, and content).
Response 6: The company (Dr. Schär, Italy) is already given in line 107.
Line 134 – change to C-reactive protein.
Response 7: We have written the “C” now in capital letter.
Line 149 – Please add the name according to Aletaha et al. [42].
Response 8: We have added the authors name as suggested, please see line 148.
Figure 1 contains STOOL on week 8. Is the data presented? Figure 1 contains QUESTIONNAIRE (HAQ, SF-36, MFI, etc). Is the data presented? Figure 1 - Is the data of cytokine measurements presented? Figure 1 - Is the data of PBMC analysis presented?
Response 9: Stool data, cytokine measurements and PBMC data are currently under investigation, bio samples were collected and stored till the end of the intervention to analyze them in parallel.
To make it clear, we would like to suggest to update the title of the article to:
MCT-Induced Ketosis and Fiber in Rheumatoid Arthritis (MIKARA) – study protocol and primary end point results of the double-blind randomized controlled intervention study indicating effects on disease activity in RA patients
Results
Line 201 – define RF and ACPA
Response 10: We have added the definitions for RF (Rheumatoid factor) and ACPA (anti-citrullinated protein antibodies), please see line 200 and 201.
Table 1. Delete the % in 18 (67%)
Response 11: since the 18 individuals were 67% of the participants in the group, we left the % sign, however it was missing in the “Test group” column, therefore we have added it here (23 (68%).
Delete Table 2. It is already explained in the text.
Response 12: Table 2 is displaying the primary endpoint results. So, we suggest to keep the table and delete the text.
Why the authors did not present the results from the secondary points from Table S2?
Response 13: Table S2 is a description of all endpoints, which are going to be investigated. This is only a first overview. Data are under investigation.
Please include the following papers and discuss their results too:
Farzana Athirah Abdul Latif, Wan Syamimee Wan Ghazali, Siti Mardhiana Mohamad, Lai Kuan Lee, High fiber multigrain supplementation improved disease activity score, circulating inflammatory and oxidative stress biomarkers in rheumatoid arthritis (RA) patients: A randomized human clinical trial, Journal of Functional Foods, Volume 100, 2023, 105392, https://doi.org/10.1016/j.jff.2022.105392.
Response 14: We have included this article, please see line 338.
Tański W, Świątoniowska-Lonc N, Tabin M, Jankowska-Polańska B. The Relationship between Fatty Acids and the Development, Course and Treatment of Rheumatoid Arthritis. Nutrients. 2022 Feb 28;14(5):1030. doi: 10.3390/nu14051030.
Response 15: We have included this article, please see line 319.

Reviewer 2 Report
The reviewer appreciates the immense efforts of the authors especially for the patient sample collections. However, the author needs to add more information about the patient history (can be added to supplementary file) especially when Glucocorticoids, other DMARDs, MTX and Biologics were administered to the patient. Is it before/on 8th week?
Author Response
Dear Reviewer,
First, thank you very much for this helpful comment, which is truly very important.
Response: Data will be summarized in supplementary materials (see table S3). Medication was not changed from T0 to T2 and to T4 (see table S3).
Amendment in the article was done in line 203.
__________________________________________________
Thank you for your second review in advance.
With best wishes,
Christina and all Authors

Round 2
Reviewer 1 Report
The authors answered properly.
Reviewer 2 Report
The reviewer appreciates the authors comments and inclusion of Table 3 in the supplementary file. It will be helpful to readers for sure. Since, the study carries significant findings; thus, the reviewer likes to recommend accepting the manuscript for publication.
However, here, the reviewer would like to suggest the authors to be careful in future while stratifying the patients into groups, especially control patients for comparison. MTX is a strong immunosuppressant and dosages variation make differences statistically.